# Comparison of Hydrogels for the Development of Well-Defined 3D Cancer Models of Breast Cancer and Melanoma

**DOI:** 10.3390/cancers12082320

**Published:** 2020-08-17

**Authors:** Rafael Schmid, Sonja K. Schmidt, Jonas Hazur, Rainer Detsch, Evelyn Maurer, Aldo R. Boccaccini, Julia Hauptstein, Jörg Teßmar, Torsten Blunk, Stefan Schrüfer, Dirk W. Schubert, Raymund E. Horch, Anja K. Bosserhoff, Andreas Arkudas, Annika Kengelbach-Weigand

**Affiliations:** 1Laboratory for Tissue-Engineering and Regenerative Medicine, Department of Plastic and Hand Surgery, University Hospital of Erlangen, Krankenhausstraße 12, 91054 Erlangen, Germany; rafael.schmid@uk-erlangen.de (R.S.); evelyn.maurer@uk-erlangen.de (E.M.); raymund.horch@uk-erlangen.de (R.E.H.); andreas.arkudas@uk-erlangen.de (A.A.); 2Institute of Biochemistry, Friedrich-Alexander University of Erlangen-Nürnberg, Fahrstraße 17, 91054 Erlangen, Germany; sonja.s.schmidt@fau.de (S.K.S.); anja.bosserhoff@fau.de (A.K.B.); 3Institute of Biomaterials, Friedrich-Alexander University of Erlangen-Nürnberg, Ulrich-Schalk-Straße 3, 91056 Erlangen, Germany; jonas.hazur@fau.de (J.H.); rainer.detsch@fau.de (R.D.); aldo.boccaccini@ww.uni-erlangen.de (A.R.B.); 4Department of Trauma, Hand, Plastic and Reconstructive Surgery, University of Würzburg, Oberdürrbacher Straße 6, 97080 Würzburg, Germany; hauptstein_j@ukw.de (J.H.); blunk_t@ukw.de (T.B.); 5Department for Functional Materials in Medicine and Dentistry, University of Würzburg, Pleicherwall 2, 97070 Würzburg, Germany; joerg.tessmar@fmz.uni-wuerzburg.de; 6Institute of Polymer Materials, Friedrich-Alexander University of Erlangen-Nürnberg, Martensstraße 7, 91058 Erlangen, Germany; stefan.schruefer@fau.de (S.S.); dirk.schubert@fau.de (D.W.S.)

**Keywords:** breast cancer, melanoma, biofabrication, hydrogel, tumor heterogeneity

## Abstract

Bioprinting offers the opportunity to fabricate precise 3D tumor models to study tumor pathophysiology and progression. However, the choice of the bioink used is important. In this study, cell behavior was studied in three mechanically and biologically different hydrogels (alginate, alginate dialdehyde crosslinked with gelatin (ADA–GEL), and thiol-modified hyaluronan (HA-SH crosslinked with PEGDA)) with cells from breast cancer (MDA-MB-231 and MCF-7) and melanoma (Mel Im and MV3), by analyzing survival, growth, and the amount of metabolically active, living cells via WST-8 labeling. Material characteristics were analyzed by dynamic mechanical analysis. Cell lines revealed significantly increased cell numbers in low-percentage alginate and HA-SH from day 1 to 14, while only Mel Im also revealed an increase in ADA–GEL. MCF-7 showed a preference for 1% alginate. Melanoma cells tended to proliferate better in ADA–GEL and HA-SH than mammary carcinoma cells. In 1% alginate, breast cancer cells showed equally good proliferation compared to melanoma cell lines. A smaller area was colonized in high-percentage alginate-based hydrogels. Moreover, 3% alginate was the stiffest material, and 2.5% ADA–GEL was the softest material. The other hydrogels were in the same range in between. Therefore, cellular responses were not only stiffness-dependent. With 1% alginate and HA-SH, we identified matrices that enable proliferation of all tested tumor cell lines while maintaining expected tumor heterogeneity. By adapting hydrogels, differences could be accentuated. This opens up the possibility of understanding and analyzing tumor heterogeneity by biofabrication.

## 1. Introduction

Various types of cancers become an increasing problem in our aging society. Nowadays, one out of eight women in the USA develops breast cancer and approximately 3% of non-Hispanic white people develop invasive melanoma [1]. Interestingly, not solid primary tumors, but their metastases, account for 90% of cancer-associated deaths [2]. Melanoma and breast cancer are the tumors with the highest frequency of metastases [3]. Different cancers vary greatly in gene expression and phenotype but also share some basic characteristics, described as the hallmarks of cancer [4,5], which can be targets for therapies. Unfortunately, only 5% of anticancer drugs that have been tested in preclinical trials successfully reach approval [6]. Apparently, most of the models in these trials do not mimic the pathophysiology properly and have to be improved to reduce failure rates. Although two-dimensional (2D) cell culture has been used for decades in research and led to substantial progress and knowledge, it has several disadvantages. The stiff plastic surface of commonly used dishes and flasks barely reflects in vivo conditions. Three-dimensional (3D) tumor models are needed for basic and applied research on tumor progression, drug efficacy, and development of resistance. There are many variables to consider when creating experimental models that are functional, reliable, and reproducible, as the tumor microenvironment consists of a complex extracellular matrix (ECM), tumor-associated cells, vasculature, and a variety of cytokines [7]. Surrounding cells can have a significant stimulatory effect on tumor cells [8]. Controlling and understanding the tumor microenvironment is necessary to understand tumor development and heterogeneity. A promising approach to create highly defined 3D constructs is 3D bioprinting using suitable bioinks [9]. With this, it is possible to arrange varying matrix and cell conditions. Such 3D models would enable the tumor cells to adopt different phenotypes that mimic the in vivo situation appropriately [10]. Several cell lines are widely used in research and may have different responses and requirements to the microenvironment. For example, the ECM between different tumors varies greatly; however, there are also many common features, e.g., in both melanoma and breast cancer, laminin 5-γ2 chain and hyaluronic acid (hyaluronan, HA) are highly expressed [11].

Over the years, many different natural and synthetic scaffold materials have been developed and used in 3D in vitro and in vivo models. HA is a naturally occurring glycosaminoglycan, consisting of glucuronic acid and N-acetylglucosamine, and it is one of the main components of the ECM with, for example, approximately 400–500 µg per gram skin [12]. HA can interact with surface receptors like CD44 and supports tissue homeostasis. It can be degraded and fragmented by mammalian cells via hyaluronidases (HYAL) during injury or remodeling and subsequently supports cell proliferation, migration, and adhesion, as well as inflammatory processes [13]. In breast cancer, the balance of HA synthesis by hyaluronan synthases (HAS) and degradation often is altered, and HA turnover can be accelerated [14]. HYAL1 expression is associated with malignant behavior of breast cancer cell lines [15]. In melanoma and the surrounding stroma, HA content is high and increased in early stages, while it is reduced in late stages [16]. HA with a high molecular weight predominates in normal tissue [17]. Different molecular weights of HA themselves can alter cancer cell behavior and chemotherapy resistance, e.g., by inducing epithelial to mesenchymal transition and cancer stem cell renewal [18]. Recently, easy-to-handle thiol-modified HAs (HA-SH) were developed for the application in tissue engineering. After the addition of a crosslinker, they form hydrogels that allow cells to grow without cell attachment [19]. They proved to be an option, for example, for breast cancer models with significant effects on the cells’ gene expression [20].

Apart from HA-based tumor models that have to be highly defined for consistent results due to possible pro-carcinogenic effects, more inert hydrogels have been established for mammalian cells. Alginate is a polysaccharide of guluronic and mannuronic acid from cell walls of brown algae, providing them with stability and flexibility. Due to its ability to be easily crosslinked and form hydrogels after adding divalent ions like Ca^2+^, it is easy to handle and frequently used for 3D models. However, mammalian cells cannot adhere to or degrade alginate. Material scientists modify the molecular structure by, for example, oxidizing alginate to alginate dialdehyde (ADA) or blending it with degradable proteins, like fibrin or gelatin, to optimize the material for 3D culture [21,22]. Additionally, it is possible to bind free amino groups like from gelatin to the aldehyde groups of the polysaccharide through Schiff’s base formation (ADA–GEL) [23], leading to advantages in matrix remodeling for mammalian cells. This approach has already been successfully used in in vitro and in vivo studies [24]. After purification, these materials are known to be biocompatible and valuable candidates for tissue engineering [24,25].

In this study, we exemplarily compared two mammary carcinoma cell lines (MDA-MB-231 and MCF-7) and two melanoma cell lines (Mel Im and MV3), to identify suitable hydrogels with different mechanical and biological characteristics (HA, alginate, or ADA–GEL) for 3D bioprinting. These cell lines are of different subtype and have different mutations to represent a range of diverse cancers with possibly varying requirements to the matrix. Material characteristics were analyzed via dynamic mechanical analysis (DMA). We focused on the effects of the hydrogels on cell survival, metabolic activity, and growth, comparing tumor cells of different origin and different subtypes.

## 2. Results

In this study, we compared the cell growth, survival, and metabolic activity of two mammary carcinoma cell lines (MDA-MB-231 and MCF-7) and two melanoma cell lines (Mel Im and MV3) in three different hydrogels, using two different hydrogel concentrations each. Alginate was used in 1% *m*/*v* and 3% *m*/*v*; ADA–GEL was used in 2.5% *m*/*v* and 4% *m*/*v*; HA-SH was used with 0.8% *m*/*v* Glycosil crosslinked with 0.5% *m*/*v* PEGDA and 1% *m*/*v* PEGDA.

### 2.1. Colony Formation and Anchorage-Independent Growth

To demonstrate typical colony forming behavior of tumor cells in standardized assays, all cell lines (MDA-MB-231, MCF-7, Mel Im, and MV3) were cultivated both in agar and Matrigel (Figure 1). Except for MCF7, the cell lines spread and formed protrusions in Matrigel. They were able to remodel the matrix. Therefore, cells infiltrated the surrounding matrix, resulting in less compact colonies in Matrigel, compared to agar. As the cells strongly proliferated and degraded the Matrigel, seven days was chosen as time point of analysis for this gel. While MCF-7 colonies were rather small with fewer cells in Matrigel, MDA-MB-231, Mel Im, and MV3 formed multicellular 3D structures with elongated cells. Over the course of 14 days, MDA-MB-231, Mel Im, and MV3 formed colonies with diameters of 100 µm and more in agar. Likewise, in Matrigel, MCF-7 colonies were also smaller, and the cells’ proliferation rate was slower compared to other cell lines. The surface of MDA-MB-231 colonies was relatively even. By contrast, MV3 and Mel Im formed colonies with cells loosely attached on the outside of the colonies and even single cells detaching. Some MCF-7 colonies spontaneously started to form mammary-gland-like structures in the agar and the Matrigel. They formed a lumen surrounded by cells, as seen on the picture in agar. Comparing these standard matrices, strong differences in cell growth were observable, based, for example, on cellular adhesion and matrix remodeling.

### 2.2. Hydrogel Properties

DMA measurements were performed to correlate the cellular behavior with the material properties (Figure 2). The storage modulus, E′ (analogous to G′ in shear rheological measurements), characterizes the pure elastic material properties. The loss modulus, E″ (G″ in shear rheology), in contrast, displays the pure viscous material properties. Since hydrogels generally do not show pure elastic or pure viscous material behavior, the value of the respective complex modulus, |E*| (or |G*| in shear rheology), was calculated. This complex modulus contains both the storage and the loss modulus.

Overall, 3% alginate was the stiffest material, followed by the other conditions. Moreover, 1% alginate, 4% ADA–GEL, and the two HA-SH (0.5% PEGDA and 1% PEGDA) gels are all in the same range, in terms of stiffness. The softest material was 2.5% ADA–GEL. The graphs of the storage modulus, E′ (Figure 2a and the complex modulus, |E*|, (Figure 2b) are relatively similar for the different angular frequencies). Therefore, the storage modulus is much greater than the loss modulus (E′»E″) for all the materials.

### 2.3. Metabolic Activity

The metabolic activity of all cell lines was analyzed over 14 days with a WST-8 assay (Figure 3), at time points day (d)1, d7, and d14. To compare the basal cell metabolic activity and survival after cell seeding between cell lines, WST-8 was performed on day one. The metabolic activity of MDA-MB-231, MCF-7, and Mel Im on day one was comparable without any significant differences. However, MV3 cells always tended to have a higher basal metabolic activity, as compared to the other cell lines. In 2.5% ADA–GEL, MV3 cells displayed up to doubled metabolic activity, which was not observed with the other cell lines.

Interestingly, both melanoma cell lines showed a trend to higher metabolic activity in ADA–GEL and HA-SH than mammary carcinoma cells during long-term cultivation (>7 days). This effect was even stronger in higher concentrated or higher crosslinked hydrogels. Both melanoma cell lines had a significantly higher metabolic activity than both mammary carcinoma cell lines in HA-SH with 1% PEGDA. In contrast, in alginate, there was a tendency for a higher metabolic activity of breast cancer cells, as compared to melanoma cells. When increasing the concentration of alginate from 1% to 3%, however, this effect disappeared—all cell lines showed similar behavior in 3% alginate.

Measured metabolic activity of all cell lines as a surrogate marker for cell proliferation increased significantly over 14 days, in 1% alginate and both concentrations of HA-SH, indicating a significant increase in cell number. Compared to 1% alginate, 3% alginate resulted in lower metabolic activity, in general. In contrast to alginate, both concentrations of ADA–GEL only led to a significant increase in metabolic activity in Mel Im. For MDA-MB-231 and MV3, there was a trend for an increase in cell metabolic activity over time in ADA–GEL, while MCF-7 even showed a decrease in metabolic activity over 14 days in this hydrogel. Similar to alginate, the low-concentrated 2.5% ADA–GEL resulted in stronger absorption values, as compared to the higher concentration of 4% ADA–GEL.

Overall, the mean WST-8 absorption of all cell lines on day 14 correlates (Figure 3b) with the mean storage modulus, E′, at 1 rad s^−1^ of each hydrogel with a Spearman correlation coefficient of ρ = −0.27. The individual cell lines showed distinct correlations. MDA-MB-231 had ρ = −0.6, MCF-7 had ρ = 0,34, Mel Im had ρ = −0.71, and MV3 had ρ = −0.26. No coefficients were significant.

### 2.4. Cell Survival and Colony Growth

With the help of Hoechst 33342, FITC-annexin V, and ethidium homodimer III, it is possible to distinguish between healthy (blue), apoptotic (green), and necrotic (red) cells by fluorescence microscopy. In all conditions, sporadic cell doublets of all cell lines were observed on day one, although they were plated from a single cell suspension. This is an indication for the ongoing mitotic activity within the gels. Cells were further stained and imaged at time points 7 and 14 days, to compare proliferation and survival over time. Appendix A shows exemplary pictures of the individual channels.

Both mammary and melanoma cell lines survived and proliferated over 14 days in alginate (Figure 4), ADA–GEL (Figure 5), and HA-SH (Figure 6), although to different extents. Sporadic cell doublets were visible in all conditions on day one. In 1% alginate and 2.5% ADA–GEL, colonies formed, which were clearly visible from day seven on. Over 14 days, they reached diameters of 80–100 µm. The colonies in alginate had mainly spherical or ellipsoid shape, while their shape was less regular in ADA–GEL, indicating that the cell lines seem to be able to remodel this hydrogel mixture. More and bigger colonies formed in 1% alginate and 2.5% ADA–GEL by all cell lines, compared to the denser gel variants. In HA-SH, spherical colonies of all cell lines were visible from day seven on. Over 14 days, cells formed mainly spherical colonies of 100 µm and more in diameter. Furthermore, compared to alginate and ADA–GEL, HA-SH facilitated fast initial colony growth. The colonies in HA-SH tended to be bigger than in alginate and ADA–GEL on day seven. There was no apparent difference between the two PEGDA concentrations in terms of colony growth or cell survival.

In 1% alginate, some channels full of cells (Figure 7a), aligned from the center to the outside, could be observed. This phenomenon was observed with all cell lines and only in 1% alginate.

All cell lines were able to escape the hydrogels and formed colonies on the bottom of the wells over 14 days. We observed this earlier in the thinner gels. An example of a colony of Mel Im bursting through the surface of a hydrogel is shown in Figure 7b.

Additionally, few MCF-7 started to differentiate and to form mammary-gland-like structures in the different hydrogels (Figure 7c). Those formed lumen surrounded by cells.

For analysis of survival after cell seeding in different hydrogels, cells were stained and evaluated at day one (Figure 8a). Generally, apoptosis and necrosis rates were relatively low for all the different hydrogels. In HA-SH gels, necrosis intensities were higher than in alginate-based gels. Compared to apoptotic intensities, necrotic rates intensities were generally higher in alginate, while the opposite effect was observed in ADA–GEL. MCF-7 showed an overall trend for the highest apoptotic and necrotic rates.

Apoptosis rates on day one were relatively low in 1% alginate for all cell lines, with no significant difference between melanoma and mammary carcinoma cell lines. Necrosis rates tended to be a little higher, especially for MCF-7. This effect was pronounced when increasing the concentration to 3% alginate; MCF-7 had a significantly higher necrotic rate than the other cell lines. MDA-MB-231 also showed a significantly higher necrotic rate than both melanoma cell lines. MCF-7 had a trend for the highest apoptotic rates in 2.5% ADA–GEL, compared to the other cell lines. Here, MCF-7 had a significantly higher necrotic rate. In 4% ADA–GEL and in both HA-SH conditions, there were no significant differences between the cell lines.

The measurement of the colonization on day 14 (Figure 8b) revealed a similar picture. Moreover, 3% alginate resulted in a smaller colonized area than 1% alginate. Both breast cancer cell lines, MCF-7 and MDA-MB-231, had a significantly higher colonized area in 1% alginate than in 3% alginate. In melanoma cell lines, this effect was not significant. There were no significant differences between the cell lines within one gel condition. In contrast, both melanoma cell lines showed a significantly larger colonized area in the 2.5% ADA–GEL, as compared to the 4% ADA–GEL. Both mammary carcinoma cell lines showed the same trend. Further, Mel Im had the significantly largest covered area in 2.5% ADA–GEL, as compared to the other cell lines. MCF-7 had the significantly smallest covered area in both ADA–GEL conditions. In HA-SH gels, there was a trend for Mel Im colonizing the largest area in both conditions. There were no significant differences between the groups.

## 3. Discussion

With this study, we compared the functional characteristics, like growth and survival, from cell lines from breast cancer and melanoma in 3D hydrogels. According to the known tumor heterogeneity, we expected differences between cells of different origin, especially in 3D culture (alginate, ADA–GEL, and HA-SH). It was the aim of this study to find a suitable material that enables cultivation and the subsequent use for bioprinting of reproducible 3D tumor models, while maintaining tumor heterogeneity.

The cell lines of mammary carcinoma (MDA-MB-231 and MCF-7) and melanoma (Mel Im and MV3) are all of metastatic origin. The breast cancer cell lines are well-established models in many different laboratories. MDA-MB-231 is a highly invasive triple-negative breast cancer (TNBC) cell line, i.e., the cells are negative for hormone receptors (estrogen and progesterone) and HER2 making it more difficult to target them with therapies [26]. MCF-7 is a less invasive luminal cell line that is positive for estrogen receptors but negative for HER2 [26], making it an easier target, and it has a heterogeneity for progesterone receptors [27]. Both have an epithelial morphology in 2D culture. The metastatic melanoma cell line Mel Im originates from a metastasis of cutaneous malignant melanoma of nodular type [28], the most aggressive type with early metastases. MV3 was also a metastasis of a nodular malignant melanoma and highly metastatic itself [29]. A mutation of BRAF in the RAS–RAF–MEK–ERK–MAP kinase pathway can be found in 66% of malignant melanomas [30]. While Mel Im has a mutation of BRAF [31], MV3 carries no BRAF mutation [32]. Therefore, the tested cell lines are very heterogeneous. Interestingly, in all our analyses, we could not only see differences between tumor types but also between the cell lines of one tumor type—a sign for tumor heterogeneity with regard to the requirements to the hydrogel.

We used Matrigel and agar as controls, to identify whether the cell lines are suitable for our model. Matrigel is a widely used hydrogel for various tumor models, since it can mimic, in part, the tumor microenvironment. However, its complex, varying, and ill-defined composition makes it difficult to use for many applications with specific questions. Matrigel is a basal membrane matrix composition from mouse sarcoma cells that is rich in growth factors, type IV collagen, laminin, entactin, and heparan sulfate proteoglycans. Therefore, they provide anchorage points that allow the cells to attach easily, while agar does not provide adhesion motifs for mammalian cells. All the cell lines were able to proliferate under both conditions (Figure 1). Hence, these cell lines have the possibility to proliferate without adhesion to the matrix and were used for our experiments. In Matrigel, the cells were able to form spreading structures that pervaded the matrix after seven days, with MCF-7 as an exception. MCF-7’s colonies were smaller and the cells proliferated less. Other groups reported a higher invasion of MDA-MB-231 in Matrigel, compared to MCF-7 [33]. Mel Im and MV3 also have been shown to have invasive properties through Matrigel [34,35]. Moreover, in agar, MCF-7 had smaller colonies than the other cell lines. MDA-MB-231’s colonies were smooth and well-defined. In contrast, MV3 formed colonies where cells detached from the aggregate. Mel Im and MCF-7 colonies had a rough surface. In these two experiments, heterogeneity was observed between all four cell lines.

Like agar, alginate also does not provide motifs for cell attachment. HA-SH (Glycosil) provides only binding for CD44 but no classic RGD sequences for integrin binding. CD44 binding can be reduced due to the thiol modification [36], depending of the substitution rate, but the presence of some CD44 binding significantly alters cellular behavior [37]. Generally, colonies in alginate and HA-SH gels were almost spherical or at maximum ellipsoid (Figure 4 and Figure 6). This indicates that the matrices are relatively homogeneous on a microscopic scale and provide a more or less even elasticity and stiffness. The gelatin in combination with the ADA (Figure 5) offers the possibility for cell adhesion through integrin binding and makes matrix remodeling easier for mammalian cells, e.g., through MMP-2 [38], which is often overexpressed in nodular melanoma and breast cancer with less favorable prognosis [39,40]. The overall stability of all hydrogels was good, and the gels were stable for 14 days. Alginate and ADA–GEL hydrogels have previously been shown to be relatively stable for 42 days, though degradation occurs [41]; the group of Sarker et al. showed that alginate has a higher swelling rate, and ADA–GEL shows a faster degradation rate, even without cells. Glycosil-based HA-SH gels have also been used for at least two weeks in vitro [42] and showed some swelling over time [43]. Hyaluronidases enhance the degradation process massively [44]. Macroscopically, the gel stability was not different in the beginning and the end of the experiments. Nevertheless, all cell lines were able to escape all of the hydrogels over time. Cells that grew on the well bottom could be observed in every condition. This indicates that all hydrogels are penetrable by the cells after a certain period of degradation. As alginate is not covalently crosslinked, Ca^2+^ can be washed out over time, and the gels degrade, depending on the molar mass distribution of the alginate [45]. The cells were also able to escape HA-SH hydrogels, as seen in Figure 7b, probably because a combination of hyaluronidases and high pressure of the colonies onto the gel surface. Within the gel, cells are in a spherical arrangement, and outside they form less dense colonies and eventually drop to the bottom of the well. This could be an interesting methodological approach to study migration properties or metastases. The cells’ transcriptome could differ significantly to 2D cultures or to the cells within the hydrogels. While it is also possible that a few cells on the surface during crosslinking are not washed away during the washing step and then settle down, Figure 7b clearly shows that an escape from inside the hydrogels, at least closer to the surface, is possible.

Overall, apoptotic and necrotic rates were low in the different gels. Considering the different apoptotic and necrotic stainings, MCF-7 cells seem to be the ones with the worst overall survival. Further, we observed the highest necrosis rates in HA-SH gels. In these gels, a smaller pipette was used due to different crosslinking setups, which could have led to higher shear forces while pipetting. MCF-7 cells seem to be more susceptible to shear stress. A model for circulating tumor cells also reported a non-significant trend for this with fluid shear stress [46]. Due to the higher viscosity in our model, the shear stress is potentially even higher. Interestingly, the ADA–GEL seemed to be the only material where cells tend to have relatively higher apoptosis than necrosis rates. The presence of gelatin may contribute to a less-artificial microenvironment and shield the cells from necrosis factors.

All of our materials showed mainly elastic properties (Figure 2), as the storage modulus is much larger than the loss modulus (E′»E″) for all the materials. In theory, one can compare the Young’s modulus with the storage modulus, E′ (or in this case the value of the complex modulus, |E*|) [47]. Since rheological measurements are performed inside the linear viscoelastic regime, using small deformations and in compression mode, a correlation to commonly performed compression tests is expected. However, deviations might derive from the fact that the Young’s modulus is calculated over a range of stresses and strains in a tensile or, for hydrogels commonly used, compression test, while the storage modulus, E′ (and the value of complex modulus |E*|), only resembles one single point in this range. When one compares the values of E′ at 1 rad s^−1^ with data of static Young’s moduli from literature, one can conclude a good agreement of both measurement techniques. The observed deviations can be caused by several factors, such as batch-to-batch variations, experimental variations, and different crosslinking techniques. The stiffest material in our experimental setup was 3% alginate with a mean storage modulus, E′, at 1 rad s^−1^ of 79.6 kPa, which decreased to a fifth for 1% alginate. Other groups reported a mean Young’s modulus, E, of 3.6 ± 0.5 kPa for 1% alginate, while stiffness increases 2.7- to 5.3-fold in 3% alginate up to 9.8 kPa to 19 kPa [48,49]. However, this is also dependent on the composition of the alginate, with respect to guluronic and mannuronic acid contents, which are specific for the harvested seaweed species (reviewed by Qin [50]) and the crosslinking cations [51]. Huang et al. postulated a roughly linear increase of the Young’s modulus in this range of single-digit percentages for alginate [49]. These differences to our results are also due to different test methods.

While our mean storage modulus in 4% ADA–GEL at 1 rad s^−1^ was 16.3 kPa, it reduced to less than a third in 2.5% ADA–GEL. This was the softest material in our setup, although the polymer content was higher than in 1% alginate. The combination of the oxidation of alginate and the addition of gelatin weakens the crosslinking. Others reported a Young’s modulus of 16 kPa of 7.5% ADA–GEL at room temperature [52]. In this gel, the modulus is even more temperature-dependent due to the gelation properties of gelatin and was reduced in another publication from 50 kPa to less than a tenth when the temperature was increased from room temperature to 37 °C [53]. Interestingly, breast cancer cells did not prefer 2.5% ADA–GEL, while it was the softest material.

Our HA-SH gels in both PEGDA concentrations had with 15.1 kPa (1% PEGDA) and 21.9 kPa (0.5% PEGDA) at 1 rad s^−1^ a similar storage modulus to 1% alginate and 4% ADA–GEL. When considering the SD of the measurements, the PEGDA concentration had an insignificant effect on the stiffness of the material. It should be noted that PEGDA in the concentrations used does not crosslink all thiol groups of HA-SH—the remaining groups can crosslink via disulfide bonds between molecules.

The total correlation of the WST-8 on day 14 of all cell lines to the storage modulus has a small negative linear correlation. MCF-7 stood out with a small positive correlation, while the others had a larger negative correlation. None of those were significant. This is an indicator that not only material stiffness is key to distinct cellular behavior. The chemical and biological characteristics of the hydrogels play a major role, as well, again, depending on the cell line. Even gels with similar mechanical properties showed completely different cellular behavior, especially for MCF-7.

The physiological in vivo Young’s modulus is approximately 1.8 kPa in normal breast tissue and can increase to 12 kPa in breast cancer [54]. In different indentation studies, it has been shown that human skin has a Young’s modulus of 1.1 kPa to 210 kPa (reviewed by Joodaki and Panzer [55]), and cancerous skin cancer shows a Young’s modulus of 52 ± 45 kPa [56]. This suggests that the average melanoma cell is situated in a stiffer environment than breast cancer cells and could be an explanation why these cells survive better in the denser gels. All the cells seemed to be influenced by the relatively stiff environment in 3% alginate, resulting in a lower proliferation and survival.

Apparently, MV3’s metabolic activity was positively influenced by the presence of gelatin or, less likely, ADA. The initial metabolic activity in the group “MV3 with 2.5% ADA–GEL” on day one of the assay tended to be higher than of the other cell lines. Compared to the microscopic analysis on day one, cell numbers were not noticeably higher in this group, although a few cell doublets had formed. Due to this high initial value, we could not detect a significant increase (only approximately 30% increase) in metabolic activity over 14 days. Nonetheless, metabolic throughput for MV3 tended to be still much higher in ADA–GEL, compared to breast cancer cell lines. Likewise, Mel Im showed the highest metabolic throughput in ADA–GEL, compared to the other cell lines. This effect increased in higher concentrated hydrogels. Similarly, in HA-SH, for both melanoma cell lines, a higher metabolic labeling was observed, indicating a higher cell number, compared to breast cancer cell lines, while this tendency became significant when increasing the PEGDA concentration up to 1%. By comparing these results to the in vivo situation, we can observe a surprising degree of similarity. In vivo, nodular melanoma is known for its aggressiveness and metastatic activity throughout the body [57]. In the present study, the melanoma cell lines, and especially Mel Im, showed fast growth and high metabolic activity in most of the conditions. Furthermore, the more invasive TNBC MDA-MB-231 was less affected by the different materials than the less invasive luminal MCF-7.

The presence of HA-SH itself within the hydrogels can have different effects on tumor cells. In our study, we could show that increasing the crosslinking of HA-SH by increasing the PEGDA concentration intensified the difference in tumor cell behavior. Previously, it has been shown that highly invasive cancer cell lines like the MDA-MB-231 have a high expression of *HAS2* mRNA and also HYAL2 [58], resulting in higher turnover and accumulation of HA with low molecular weight. The less invasive MCF-7 has a lower degradation rate [59]. Furthermore, it has been demonstrated that low-molecular-weight HA can promote tumor cell invasion of MDA-MB-231, while high-molecular-weight HA cannot, as high-molecular-weight HA inhibits migration [60,61]. It was shown that HAS3 overexpression in MV3 reduced cell adhesion, migration, and proliferation [62]. Therefore, altering the HA-SH could be used to study different tumor properties.

Notably, some MCF-7 cells started to spontaneously form mammary-gland-like structures in all different hydrogels (Figure 7c). The cells self-organize to form a luminal space, surrounded by non-homogeneously distributed cells. This process seems to be only possible in 3D. Here, cell–cell and cell–matrix interactions vary greatly from the standard 2D setting. As this also happens in the anchorage-independent growth in agar, one can assume that the cell interaction in 3D is key to the differentiation. Others reported this phenomenon in agarose [63]. The heterogeneous differentiation, that only a fraction of our colonies shows, suggests again that the 3D models are closer to the in vivo situation than the 2D cell culture. Additionally, we can better adapt 3D models to specific in vivo situations and the needs of the cells.

In summary, the biggest advantages of the model presented in this study are the following. In contrast to other hydrogels, like Matrigel, the gels of this study are well-defined and tailorable, e.g., in terms of stiffness and growth factors. Therefore, they can be used to answer questions that are more sophisticated, also while accentuating tumor heterogeneity. Using extrusion-based 3D printing, we can further improve the models, e.g., with the implementation of stiffness gradients, the addition of other cells or stimulating factors in a defined spatial organization. In addition, a macroporous structure could ensure homogeneous crosslinking, which could be difficult for alginate-based bioinks otherwise. Thicker constructs may have low diffusion rates. Variations of the different hydrogels in this study have already been used for extrusion-based 3D printing. Although alginate itself has poor printing properties due to low shape fidelity, a pre-crosslinking step with CaCO_3_ and D-Glucono-δ-lactone can significantly improve the printing results without increasing the polymer density [64]. Due to the gelation properties of gelatin, ADA–GEL has also been shown to be printable [65]. Bioprinted HA-SH gels have also been produced by using, for example, the photo initiator Irgacure 2959 [66]. Here, the printing properties seem improvable due to low shape fidelity.

This study confirms that we could successfully simulate in vivo conditions in which the different cell lines showed the expected heterogeneity, with respect to origin and subtype. We even could pronounce this effect by increasing the hydrogel concentration or crosslinking density. This makes it possible to study tumor heterogeneity further. Generally, all cells proliferated better in hydrogels of lower concentrations. In this in vitro model, various other cancer cell lines could be tested by using these hydrogels. Subsequently, it should be noted that establishing a 3D tumor model makes it necessary to carefully choose hydrogels for each cell line individually.

## 4. Materials and Methods

### 4.1. Cell Culture

All tumor cell lines in this study were of metastatic origin. MDA-MB-231 and MCF-7 (both from American Type Culture Collection ATCC, Manassas, VA, USA) were cultivated in DMEM high glucose (Thermo Fisher Scientific, Waltham, MA, USA), 2 mM L-glutamine (Sigma-Aldrich, St. Louis, MO, USA), 10% FCS Superior (standardized fetal bovine serum, Biochrom GmbH, Berlin, Germany), and non-essential amino acids (Gibco Life Technologies, Carlsbad, CA, USA). Mel Im was cultivated in DMEM low-glucose (Sigma-Aldrich) with 2 mM L-glutamine, MV3 in DMEM high-glucose (Sigma-Aldrich) with 2 mM L-glutamine, with the addition of 10% FCS each. MV3 were obtained from Peter Friedl (RIMLS Radboud University Nijmegen Medical Center, Nijmegen, The Netherlands). The media were supplemented with penicillin/streptomycin (100 U mL^−1^, 0.1 mg mL^−1^, Sigma-Aldrich), and the incubator was set to 5% CO_2_, at 37 °C.

### 4.2. Colony-Forming Assay

For the evaluation of attachment-independent growth, cells were seeded into soft agar. Six-wells were coated with 2 mL of a 0.5% base agar, which was prepared as a dilution of 2% *m*/*v* agar-agar (Carl Roth GmbH + Co. KG, Karlsruhe, Germany) in DMEM supplemented with 20% FCS (Sigma-Aldrich), 1.925 mg mL^−1^ sodium bicarbonate, and 1× non-essential amino acids. Then, 5 × 10^3^ cells were seeded in triplicates, into 1 mL 0.3% *m*/*v* top agar (diluted from base agar composition with cell suspension in DMEM) per well. The plates were incubated for 14 days, and the colonies were observed under an inverted microscope (Olympus IX83, cellSens Software V1.16, Olympus Corporation, Tokyo, Japan).

The capability to form colonies from single cells was also evaluated by using Matrigel (Corning Inc., Corning, NY, USA). Then, 2 × 10^4^ cells were suspended per mL Matrigel, and 100 µL was seeded per 96-well, in triplicates, and incubated at 37 °C, for 30 min. Afterward, 100 µL of medium was added. Pictures were taken at day 7, using an inverted microscope.

### 4.3. Hydrogels for 3D Cell Culture

For alginate hydrogels 1% *m*/*v* respectively 3% *m*/*v* VIVAPHARM^®^ Alginate PH 176 (JRS PHARMA GmbH & Co. KG, Rosenberg, Germany) was dissolved in PBS without Ca^2+^ and Mg^2+^ (Sigma-Aldrich). A positive displacement pipette was used to resuspend the cells (1 × 10^6^ mL^−1^) and form hydrogel beads by pipetting droplets of approximately 100–150 µL (3–4 mm in diameter) of the cell suspension into a CaCl_2_ bath (100 mM, Sigma-Aldrich). After 10 min of crosslinking, the beads were washed by using 100 mM HEPES (Sigma-Aldrich), for 10 min, and then transferred into cell culture medium and incubated at 37 °C and 5% CO_2_, for 14 days. Medium was changed three times per week.

ADA was synthesized by using VIVAPHARM^®^ Alginate PH 163 S2 (JRS PHARMA GmbH & Co. KG), based on previous experiments [23]. Briefly, 10 g of alginate was dispersed in 50 mL of ethanol (Sigma-Aldrich), and 50 mL of a 125 mM aqueous NaIO_4_ solution (Sigma-Aldrich) was added slowly to the stirring dispersion, in the dark. After 6 h, 10 mL of ethylene glycol was added to quench the reaction. After 30 min, most of the supernatant was removed, and the residual ADA filled into dialysis tubes (MWCO 6–8 kDa, Repligen, Waltham, MA, USA) and dialyzed for 3–4 days. Afterward, the solution was lyophilized, using an Alpha 2-4 LSCplus (Martin Christ Gefriertrocknungsanlagen GmbH, Osterode am Harz, Germany).

Then, 5% *m*/*v* respectively 8% *m*/*v* ADA was dissolved in PBS, without Ca^2+^ and Mg^2+^, and 5% *m*/*v* respectively 8% *m*/*v* solutions of porcine gelatin (Sigma-Aldrich) were prepared. The two solutions of the same concentration were mixed and crosslinked to ADA–GEL through stirring at 37 °C for 10 min. ADA–GEL beads were produced as described for alginate beads.

Hyaluronic-acid-based hydrogels were produced by using the HA-SH Glycosil, and polyethylene glycol diacrylate (PEGDA, Mw = 3,350 Da) (both ESI BIO, Alameda, CA, USA). Glycosil was used in a final concentration of 0.8% *m*/*v* and PEGDA in a concentration of 0.5% *m*/*v* or 1% *m*/*v*. Cells were mixed with the components (1 × 10^6^ mL^−1^) and 40 µL pipetted into glass cloning cylinders with an outer diameter of 6 mm and an inner diameter of 4 mm. After 30 min incubation in the incubator at 37 °C for the gelation process that utilizes the Michael addition, a piston was used to transfer the hydrogels into cell culture medium. Hydrogels were cultivated at 37 °C, at 5% CO_2_, for 14 days. Medium was changed three times per week.

### 4.4. Apoptotic/Necrotic/Healthy Staining

The apoptotic/necrotic/healthy staining (PromoCell GmbH, Heidelberg, Germany) was performed on days 1, 7, and 14. In short, hydrogels were washed with the kit’s binding buffer and incubated for 15 min in the staining solution containing FITC-annexin V (for apoptosis), ethidium homodimer III (for necrosis), and Hoechst 33342 (for DNA). Next, hydrogels were washed once again and evaluated, using an inverted fluorescence microscope. Cells that emit solely blue fluorescence were considered healthy, red fluorescence necrotic, and green fluorescence apoptotic.

For the evaluation of the cell survival on day 1, the mean color intensities of the individual channels were measured after applying a background subtraction, using the rolling ball algorithm of Fiji Is Just ImageJ (Fiji) 1.52p [67], an extended distribution of ImageJ. These values were then normalized to the exposure time of each channel and set in relation to the intensity of DAPI. Each time, three representative images, one picture per individual experiment, were analyzed.

### 4.5. Colonization

To evaluate the rate of colonization, quantification of phase-contrast images was performed on day 14, using Fiji. Pictures were taken close to the bottom of the constructs. The area of the colonies in focus within the gel was manually measured and set in relation to the visible gel area. Blurred colonies were excluded from quantification. Each time, one picture per individual experiment—in total, three representative images—were analyzed.

### 4.6. Cell Metabolic Activity Assay

WST-8 assays (PromoCell GmbH) were performed to test for the amount of metabolically active cells on days 1, 7, and 14. The hydrogel constructs were transferred into wells of a 96-well plate, in triplicates. New constructs were used each time. Then, 100 µL of medium and 10 µL of the tetrazolium salt WST-8 were added and incubated for 2 h, at 37 °C. Afterward, 100 µL was pipetted into new wells and measured by using a microplate reader (Thermo Fisher Scientific), at 450 nm, with a reference of 600 nm.

### 4.7. Dynamic Mechanical Analysis

For characterizing the viscoelastic properties of the samples, DMA tests were performed. The different gels were prepared in the same concentrations as described above. For alginate and ADA–GEL, films of 1 mL were prepared in a 12-well plate, resulting in circular samples with a diameter of 18 mm and a height of 2 mm. Alginate gels were crosslinked for 10 min in 100 mM CaCl_2_. ADA–GEL was crosslinked for 40 min. HA-SH gels were cast into 12 mm Teflon rings (226 µL per ring) and crosslinked for 30 min. The different gels were transferred into complete cell culture medium, put into the incubator at 37 °C, and measured for their mechanical characteristics, on day 1.

A DHR-3 rheometer (TA Instruments, New Castle, DE, USA), equipped with a 20 mm plate–plate geometry for alginate and ADA–GEL samples and a 12 mm plate-plate geometry for HA-SH samples in oscillating compression mode, was used for all tests. An axial pre-force of 0.1 N was applied to all samples, except for the 2.5% ADA–GEL samples, since a drastic deformation was already observable under 0.1 N axial load and was therefore reduced to 0.05 N. All tests were performed with a constant temperature of 37 °C, controlled by the attached Peltier element. An equilibration time of 3 min prior to each measurement was proven as sufficient to ensure a homogeneous temperature distribution in preliminary tests. Amplitude sweeps were performed at a frequency of 1 rad s^−1^ for determining suitable amplitudes for the following frequency sweeps. The amplitude was chosen as a trade-off between measurement noise at very low amplitudes and non-linearities, e.g., gel destruction or wall-slip, at larger amplitudes. The time and frequency dependent viscoelastic properties were determined by frequency sweeps. A range of 0.1 to 10 rad s^−1^ (0.02 to 1.5 Hz) is covered for all materials. This regime is relevant for revealing relations of mechanical properties and cell studies, since those are performed in a quasi-stationary state. Average values and standard deviations were calculated from at least 3 measurements of technical replicates. All measurements were additionally rated by the following criteria and only evaluated if all were met:The axial force recorded by the instrument did not drastically alter during the measurement.The oscillation force was significantly bigger than the lower limit given by the manufacturer.Raw data signals showed a sinusoidal function for recorded force and displacement.Both moduli (storage modulus, E′, and loss modulus, E″) were positive and recorded by the instrument.

The complex modulus |E*| was calculated as follows:
$$|E^{*}|=\sqrt{E^{\prime 2}+E^{\prime \prime 2}}$$

### 4.8. Statistics and Figures

All experiments, except for DMA, were performed three times (*n* = 3). Statistical analysis was performed by using IBM SPSS statistics software V24 (SPSS Software/IBM, Armonk, NY, USA). Differences between groups were analyzed, using the Kruskal–Wallis H test, followed by a Mann–Whitney U test for post hoc analysis; the asymptotic significance was used. Significant *p*-value was set to ≤ 0.05. Figures show the mean ± standard deviation and were created with GraphPad Prism 8.1.2 (GraphPad Software, La Jolla, CA, USA). Depicted microscopic images were arranged and edited with CorelDRAW X6 and Corel PHOTO-PAINT X6 (Corel Corporation, Ottawa, ON, Canada), to compensate differences in brightness and contrast to match the observed results.

## 5. Conclusions

With this study, we compared survival, metabolic activity, and growth of cell lines from mammary carcinoma and melanoma in different hydrogels for bioprinting (alginate, ADA–GEL, and HA-SH), with regard to tumor heterogeneity. It could be demonstrated that hydrogels have vastly differing mechanical properties and have to be carefully chosen according to the respective demands of each cell line or the experimental question to optimally mimic the in vivo condition of different tumor (sub)types. Generally, melanoma cells showed a tendency for a higher metabolic activity in ADA–GEL and HA-SH, compared to breast cancer cells, while this effect got stronger with increasing concentration of the hydrogels. In contrast, mammary carcinoma cells showed equally good proliferation in 1% alginate, as compared to melanoma cells. These different cell lines showed different characteristics in vitro that could also be observed in vivo, with respect to origin and subtype. We conclude that biofabrication of these hydrogels may offer a suitable model to study and understand tumor heterogeneity.

## Figures and Tables

**Figure 1 cancers-12-02320-f001:**
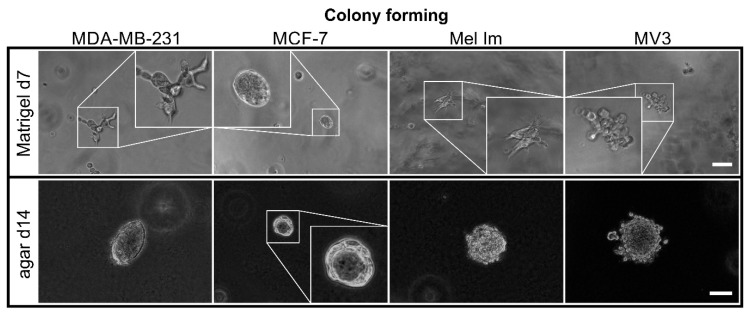
Phase contrast images of cell lines in Matrigel (top) and agar (bottom); all cell lines (MDA-MB-231, MCF-7, Mel Im, and MV3) were able to form multicellular colonies from single cells in both matrices; scale bar = 100 µm.

**Figure 2 cancers-12-02320-f002:**
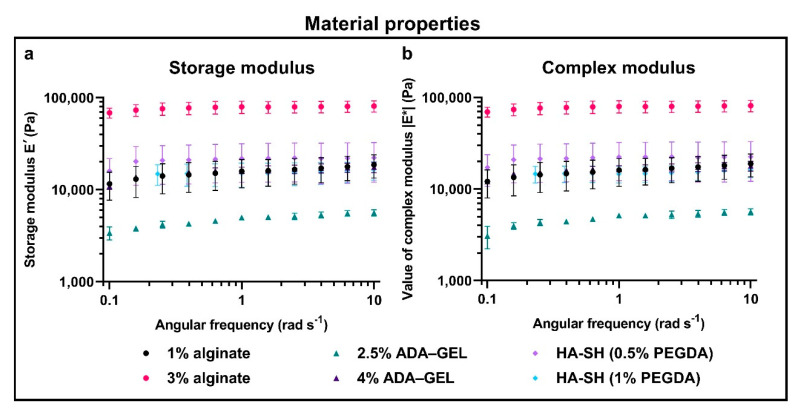
Material properties of hydrogels on day one: (**a**) the storage modulus (E′) and (**b**) the complex modulus (|E*|). Data shown as mean ± SD of technical replicates.

**Figure 3 cancers-12-02320-f003:**
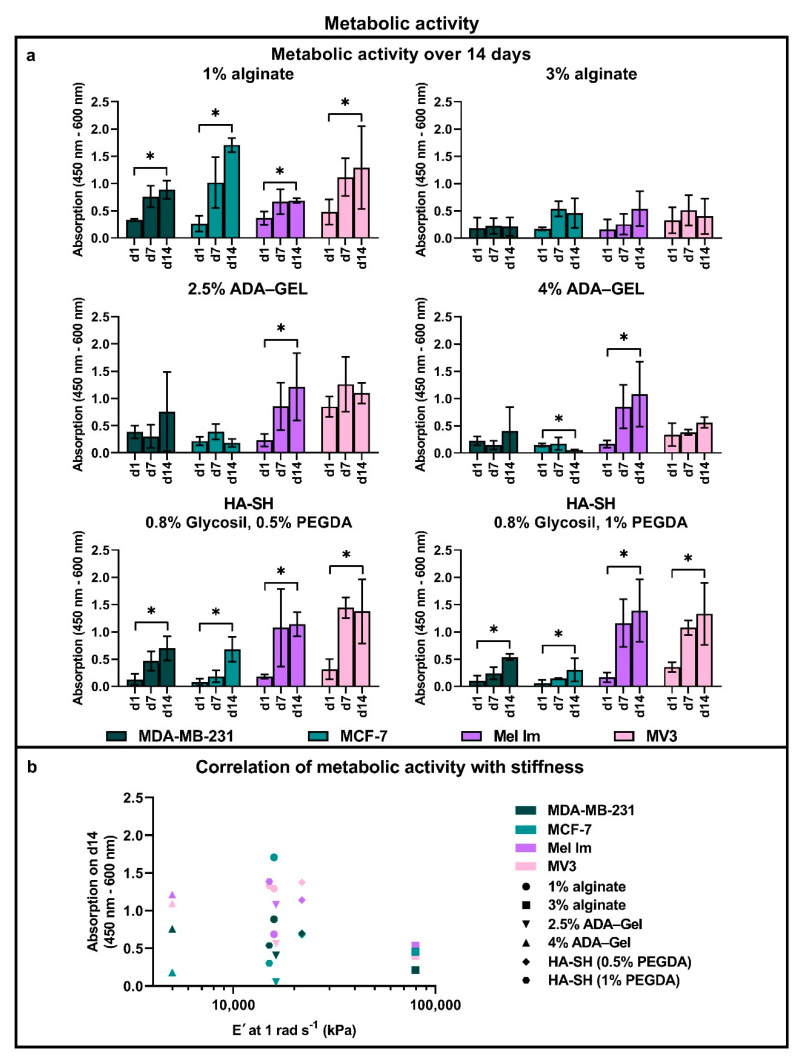
Cell metabolic activity of four cell lines (**a**) in alginate, ADA–GEL, and HA-SH, measured by using a WST-8 assay over 14 days; not all significances are shown for better clarity but mentioned in the results; data shown as mean ± SD of biological replicates, * *p* ≤ 0.05 Kruskal–Wallis H test, Mann–Whitney U test, *n* = 3; (**b**) correlation of the metabolic activity of four cell lines on day 14 with the stiffness of different hydrogels.

**Figure 4 cancers-12-02320-f004:**
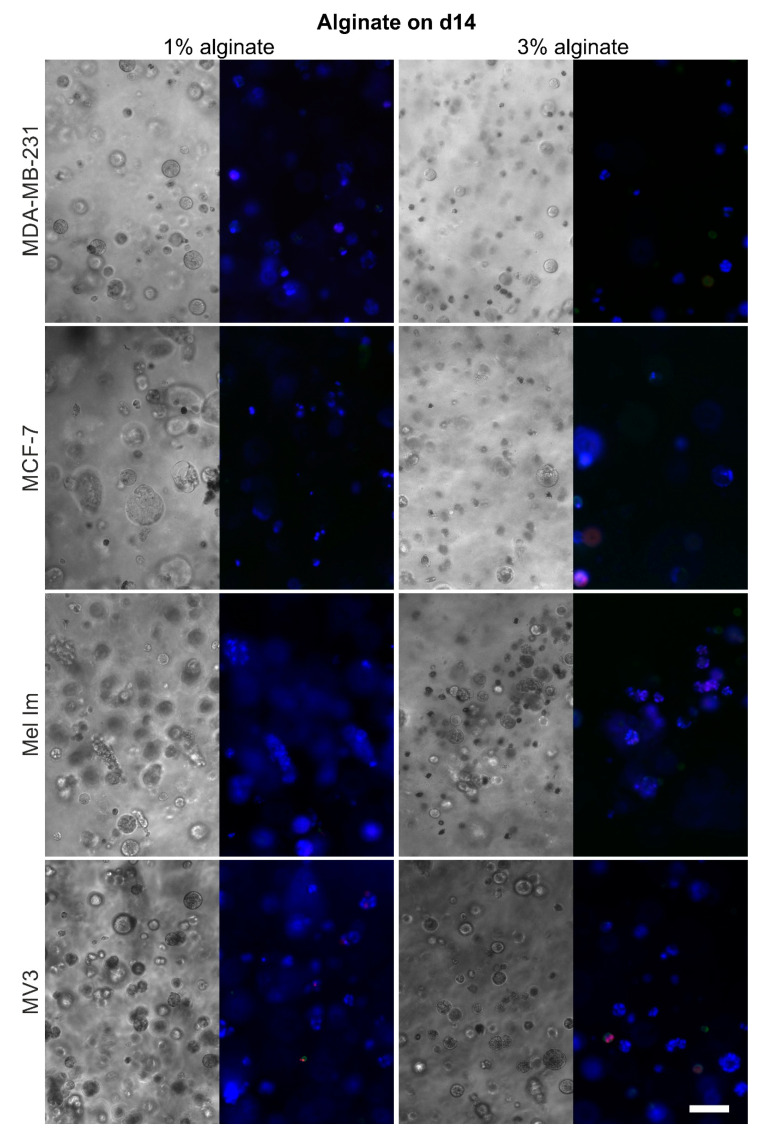
Representative phase contrast and fluorescence pictures of the cell survival and colony formation in alginate on day 14; adjacent pictures show the same position; blue = nucleic acid stain, green = apoptosis, and red = necrosis; scale bar = 100 µm.

**Figure 5 cancers-12-02320-f005:**
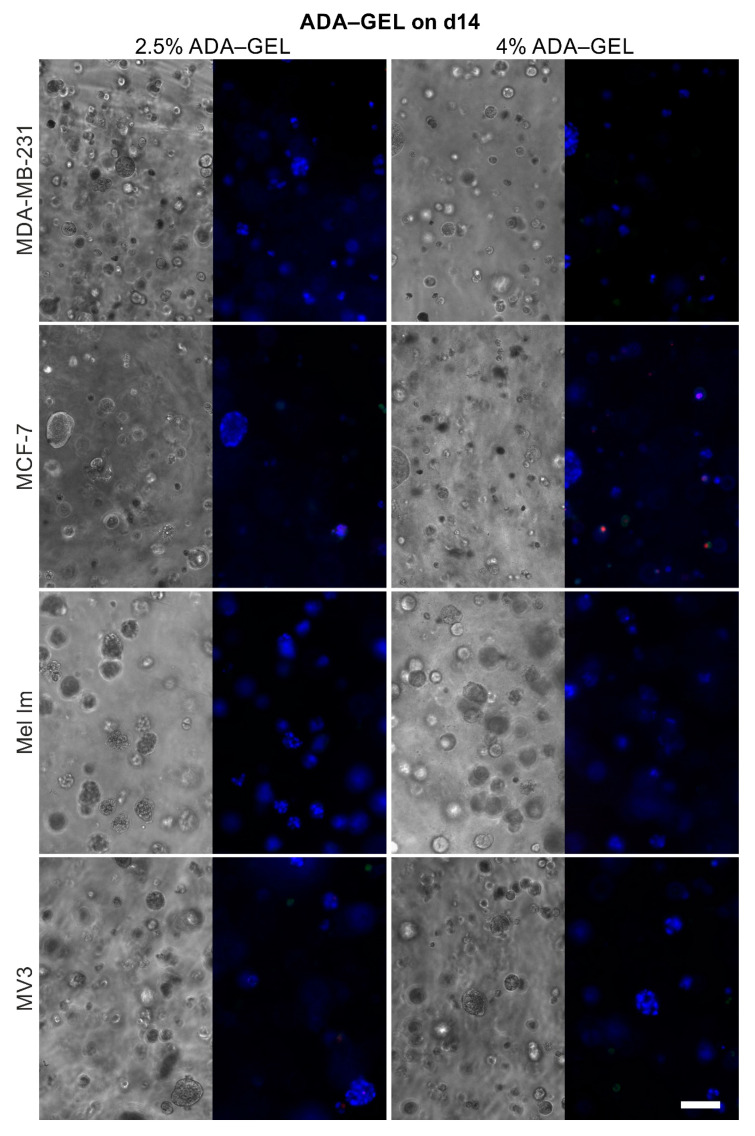
Representative phase contrast and fluorescence pictures of the cell survival and colony formation in ADA–GEL on day 14; adjacent pictures show the same position; blue = nucleic acid stain, green = apoptosis, and red = necrosis; scale bar = 100 µm.

**Figure 6 cancers-12-02320-f006:**
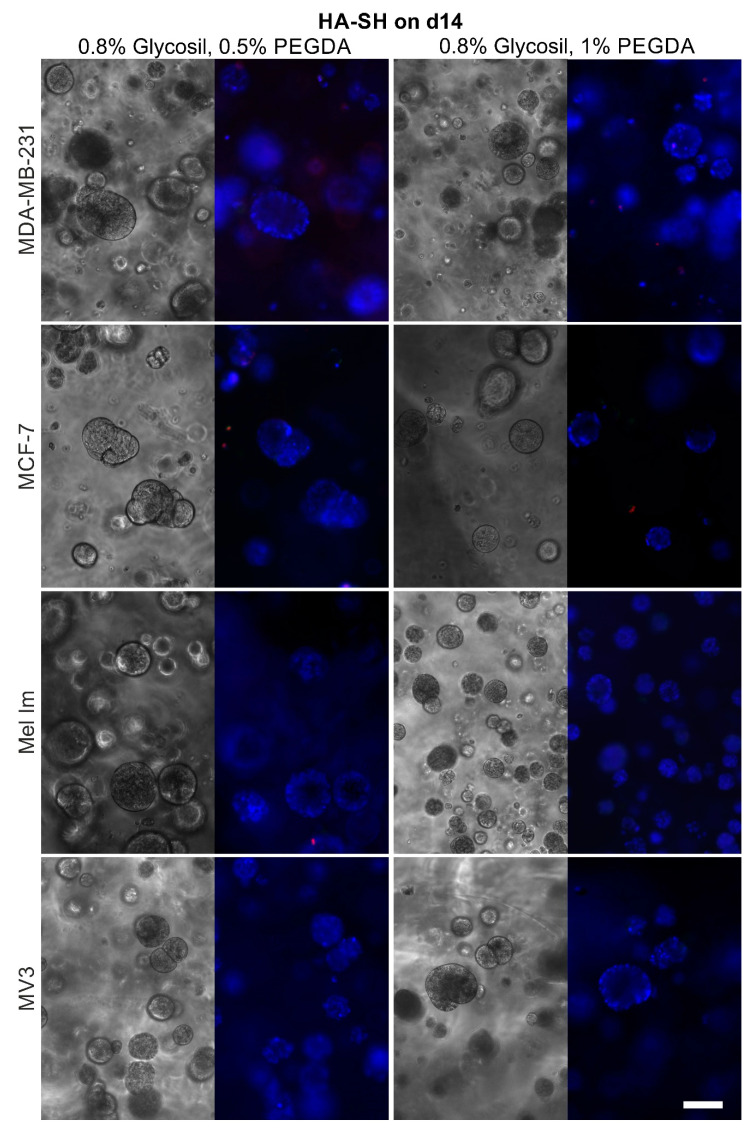
Representative phase contrast and fluorescence pictures of the cell survival and colony formation in HA-SH on day 14; adjacent pictures show the same position; blue = nucleic acid stain, green = apoptosis, and red = necrosis; scale bar = 100 µm.

**Figure 7 cancers-12-02320-f007:**
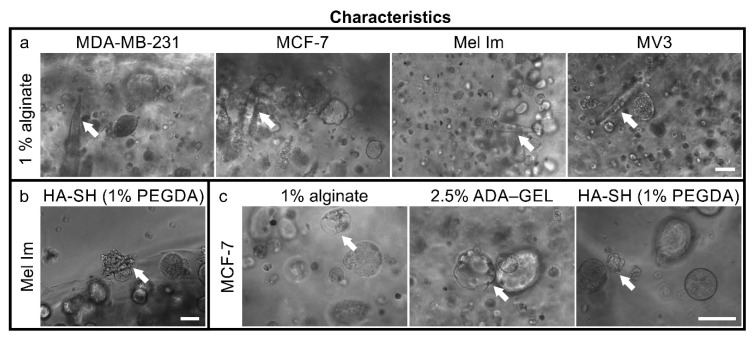
Cellular characteristics on day 14 in hydrogels: (**a**) channels of cells within 1% alginate aligned from the center to the outside, indicated by the arrows; (**b**) Mel Im that burst the surface of the hydrogel, indicated by the arrow; (**c**) formation of mammary-gland-like structures of MCF-7 in different hydrogels, indicated by the arrows; scale bars = 100 µm.

**Figure 8 cancers-12-02320-f008:**
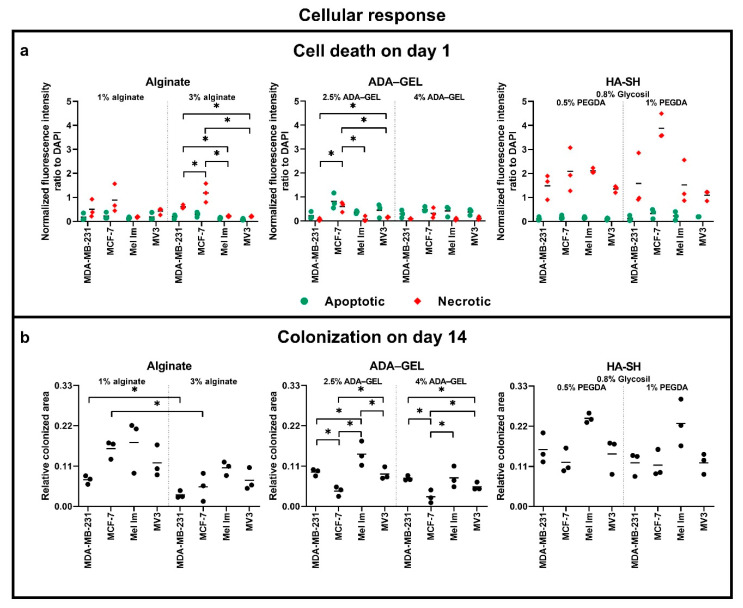
Cellular response of four cell lines in alginate, ADA–GEL, and HA-SH: (**a**) quantification of the apoptotic/necrotic/healthy staining on day one; (**b**) quantification of the colonization in phase-contrast images on day 14; * *p* ≤ 0.05 Kruskal–Wallis H test, Mann–Whitney U test, not all significances are shown for better clarity but mentioned in the results, *n* = 3 (one picture each was analyzed).

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
