# Peer review of "Comparison of Hydrogels for the Development of Well-Defined 3D Cancer Models of Breast Cancer and Melanoma"

_cancers, 2020, doi:10.3390/cancers12082320_

Round 1

Reviewer 1 Report

The authors compare and describe the different hydrogel systems for 3D cell culture using breast cancer and melanoma cell lines. They analyzed the cellular growth using different biological assays and conclude that the hydrogel system used should be chosen based on the needs of the particular cell line used. Overall this is nice work but there are some corrections that need to be made to the manuscript.

Figure 1 - The caption is incorrect. Agar images are on the bottom and Matrigel images are on the top, in contrast to what appears in the caption.

Figure 2 - It is difficult to see the points in the graphs. Can the size be increased to make this easier for the reader?

Section 3.2. The authors interchange metabolic activity and proliferation, although it is well known that this can be a flawed assumption. Metabolic activity can increase for example, when cells are stressed, or metabolic activity can decrease when cells begin to differentiate. For the most part, the authors refer to the data as metabolic activity and this is how the results should be described. The authors describe that MCF-7 cells did not proliferate in the ADA-GEL (lines 173-174), yet it is clear that the cells have proliferated from a single cell suspension when we look at Fig 6. Either the authors should change the term from 'proliferation' to 'metabolic activity' throughout the manuscript, or include a more direct assay (such as DNA content) to enhance the claims that they are observing changes in proliferation and not just metabolic activity.

Fig 2 and 3 - Is data represented as mean and standard deviation? Please include what the data in the graphs represents in the figure legends.

Lines 233-246 - It is difficult to tell from the images if there are positive cells for necrosis and apoptosis. Can the authors also include zoomd-in images in Figs 4, 6 and 7 to be able to see the fluorescence images clearly?

Fig 8 - How was the colonization determined/calculated. It is not clear from the methods how this was performed.

Discussion lines 312-313 - How do the authors know that the cells were able to degrade and escape from the hydrogels, rather than cells being present on the surface of the gel during cross-linking/gel preparation, and the cells naturally migrating from the gel surface to the attractive tissue culture plastic surface?

Author Response

Thank you very much for reviewing our manuscript. Please see the attachment.

Reviewer 2 Report

This article compare hydrogels for the development of 3D cancer models of breast cancer and melanoma. In this study, cell behavior was studied in three mechanically and biologically different hydrogels (alginate, alginate dialdehyde crosslinked with gelatin (ADA-GEL), and thiol-modified hyaluronan (HA-SH crosslinked with PEGDA)) with cells from breast cancer (MDA-MB-231, MCF-7) and melanoma (Mel Im, MV3), by analyzing survival, growth, and the amount of metabolically active, living cells via WST labeling. Material characteristics were analyzed by dynamic mechanical analysis.

The study is of interest to readers working on cancer research; however, some concerns may need to be addressed before publication. First, the authors may need to compare their model with other more extensively used models (e.g., in vivo models) for cancer. This is important because this can show how well the reported model can be used in reality. In addition, authors may need to explain why the two cancer cell type is chosen. Can the same method be used to model other types of cancer? Finally, how the viability of the cells change over time? Also, what are the limitations of the models reported? All these are areas that may need to be clarified during revision.

Author Response

(The authors gave the same response as above.)

Reviewer 3 Report

This is a well-designed and thorough study of the impact of different hydrogels on how cancer cells develop into tumors.

The manuscript needs some modification to address these relatively minor points:

The Discussion section would benefit from being shortened

Line 30  double parentheses should be corrected

32  WST labeling should be defined

55  "reach registration" should be clarify to also indicate "approval" of the drug.

60 resistances -> resistance

71 a lot common features -> many common features

127  in contrast -> by contrast

131 are abservable -> were observable (the Results section should be in the past tense

140  storage modulus and complex modulus should be explained briefly

Author Response

(The authors gave the same response as above.)

Reviewer 4 Report

The authors have prepared several 3D hydrogels with different composition and tested their functional characteristics to develop bioprinted 3D tumor models. They have focused their work on the cell survival, growth and metabolic activity of two breast cancer and two melanoma cells lines. In general, the discussion and results are well addressed, but the conclusions are not supported by the results obtained. The work requires further analyses and experiments. Those, together to some modifications and unclear questions pointed out below, might improve the quality of the manuscript:

  1. Which is the reason of forming hydrogel beads? Which is the dimension of those beads?
  2. In Figure 2, the values showed for these two moduli seem the same. Which is the physical difference between them?
  3. In page 6, authors state that cells “were able to escape the hydrogel over 14 days and formed colonies on the bottom of the wells.” Have the authors performed mobility studies of these cells or have related information about it? Maybe larger hydrogels or hydrogel-filled culture wells would avoid such “escape”. Have the authors performed similar experiments?
  4. The description of the results observed in cell survival and colony growth at point 3.3 is similar for the three kind of hydrogels. The manuscript looks repetitive as it is and makes reference to figures that are far (Figures 5b and c are mentioned in page 10, after Figure 7 and two pages after its appearance, what difficult the overall understanding). To make it clearer to the reader, I suggest reorganizing and join the information – for example, instead of describing all the characteristics of each hydrogel, describe the same observable for the different hydrogels in each paragraph.
  5. Swelling and (bio)degradation analyses are strongly recommended.
  6. According to authors discussion, all cells are able to escape the hydrogel, and it is unclear if it due to the lack of attachment of cells to them or the degradation of the gels. The authors affirm that gelatin with ADA does offer the possibility of cell adhesion through integrin binding, but they also affirm they observe cellular escapement in all the tests. Therefore, the hydrogels developed do not look like good substrates for creating tumor models, and, thus, the aim of the manuscript has not been achieved. How would the authors suggest improving such lack of adhesion and reach the objectives planned?
  7. The hydrogels are aimed for 3D bioprinting, but no 3D printing tests are presented. Which pattern and gelation protocol would the authors implement? It is strongly suggested to provide evidence of printability.
  8. Have the authors planned to develop non-printed large-size hydrogels? Cubic hydrogels of mm or even cm sizes would be interesting as tumor models and would not require the use of 3D printing.

Author Response

Thank you very much for reviewing our manuscript and your very detailed and well thought out comments. They were extremely helpful for the revision of our manuscript.

Round 2

Reviewer 1 Report

I am satisfied with the authors response to my questions.

One minor point: on line 175 the sentence seems incomplete. The authors write "...surrogate marker for cell increased significantly..." Perhaps this should read "...surrogate marker for cell proliferation increased ..."

Author Response

Thank you very much for reviewing and approving our manuscript. We corrected the error.

Reviewer 2 Report

The authors have addressed the concerns raised previously. I recommend this article to be considered for publication.

Author Response

Thank you very much for reviewing and approving our manuscript.

Reviewer 4 Report

Authors have addressed well almost all the comments. However, I am afraid that due to the mismatch in the versions provided, line labels have been moved and the new information cannot be found in the lines they say. Therefore, I cannot properly review it. Please find attached my comments to the author’s response. Some other minor comments/suggestions are added.

Author Response

Thank you very much for reviewing our manuscript once again. We are very sorry about the mismatch with the lines. Please see the attachment.
